# Bull efficiency using dairy genetic traits

**Christine E. Whitt**[1]*, **Loren W. Tauer**[2], **Heather Huson**[3]

**1** United States Department of Agriculture, Economic Research Services, Washington D.C., United States of America, **2** Charles H. Dyson School of Applied Economics and Management, Cornell SC Johnson College of Business, Cornell University, Ithaca, New York, United States of America, **3** Department of Animal Science, College of Agriculture and Life Sciences, Cornell University, Ithaca, New York, United States of America

* cs2292@cornell.edu

## Abstract

Dairy bulls are evaluated using progeny data and genomic testing to determine the quantity of specific traits that they will pass to their daughters. Some bulls excel in some traits but not others. Specifying these various traits as outputs, with the single input of insemination, technical, revenue, allocative, and profit efficiency of bulls available for artificial insemination are estimated using Free Disposal Hull. Although bulls generally are highly technically efficient, because only high performing bull semen is offered for sale, bulls are less revenue, allocative and profit efficient. These efficiencies are relative to peer bulls and can be updated as new bulls become available.

## Introduction

Efficiency of individual farms have been frequently estimated, with dairy being possibly the most studied type of agricultural commodity [1]. The efficiency performance of crops have also been estimated [2], and the role of genetics and crop breeding in productivity growth has been explored [3] and [4]. In this study we estimate the efficiency of dairy bulls given the genetic traits that they may pass on to their daughters who are retained for milking purposes. Bulls available for artificial insemination are genetically tested, with values calculated for the various traits deemed important in milk production. Performance of their daughters is later used to update these measures. Farmers can peruse the list of bull offerings and select bulls to match with cows based upon the values of the traits that each bull may transmit on to their daughter.

The USDA has derived the relative importance of each trait in milk production with an associated value, and have derived valuation indices, referred to as net merit values. The USDA constructs an overall general net merit index, but also net merit indices for producers who graze their cows rather than use confinement systems, where stature and size are important, and for farmers who produce for a cheese market, rather than fluid milk. These net merit values are the net economic value that a cow should contribute to farm profits over her lactations. To assist farmers in making decisions, the USDA publishes the quantity and economic value of each individual trait [5]. The USDA Net Merit economic weights on the various traits are: Protein (20%), Fat (22%), Milk Yield (-1%), Productive Life (19%), Somatic Cell Score (-7%), Udder Composite (8%), Feet/legs Composite (3%), Body Size Composite (-5%),

**Competing interests:** The authors have declared that no competing interests exist.

Daughter Pregnancy Rate (7%), Heifer Conception Rate (2%), Cow Conception Rate (1%) and Calving Ability Dollars (5%).

Picking the bull with the higher net merit score is tantamount to picking the bull that will produce a daughter that will produce the greatest net revenue. By dividing the net merit score of each bull by the highest net merit score, the revenue efficiency of each bull can be calculated. Revenue efficiency can be decomposed into technical and allocative efficiencies, which we do using free disposal hull (FDH). FDH is used because only one sperm impregnates a cow; no linear combination of bulls produce a daughter. By including the price of the semen into the analysis, we also compute profit efficiency.

Bulls that are not revenue or profit efficient are still offered for sale and farmers select those bulls for insemination because those bulls have traits that farmers desire to more strongly establish into their herds [6]. The price (net economic return) of a trait used to construct the USDA net merit values are based upon net return for a representative farm, but a farmer may place a higher or lower value on those traits, resulting in some bulls, even if technically efficient, not being allocative efficient, or even profit efficient.

Except for discussion in a textbook on benchmarking, where [7] discuss a project to evaluate bull genetic traits, there appears to be no other application estimating the efficiency of genetic traits, although others have estimated the value of traits in bulls. [8] used Holstein bull semen data to analyze the value of production and type characteristics. They modeled breeding decisions by expressing them as a function of the net present value of all future outcomes and included supply effects of semen pricing in a hedonic price model. They concluded that supply-side effects were significant contributors in determining the final price of semen. [9]4and [10] determined the value of genetic merits by estimating what those traits add to estimated profit. [11] also examined the potential increase in farm profits to determine the value of production characteristics. [12] used linear programming to investigate the effect of six genetic goals on minimum semen costs, sires selected for service and economic values.

[13] argued that changes in the cost curve are what determine the profit function for genetic improvement in animal breeding, and the link between costs and the profit function is key to determining the true value of genetic merits. Indeed, 5[14] estimated a translog variable cost function, along with the associated factor share equations, to determine the cost reduction of various genetic traits in a Norwegian dairy herd. In additional to the traditional inputs one would expect in a dairy cost curve, they included three genetic merit indices dealing with production, function, and conformation. The estimates were a 0.4 and 0.6 percent cost reduction on the average farm for a one standard deviation in the production and functional indices, respectively. The estimated cost coefficient estimate on the conformation index was positive rather than negative. [15] estimated an input distance function and derived Malmquist productivity indices for Icelandic dairy farms. Average productivity growth over the period 1997 to 2006 was 1.6 percent, with 19 percent of that productivity growth rate being due to breeding.

Given the unique nature of this genetic trait data, which does not represent the normal specification of outputs and inputs in measuring efficiency, we first discuss the process of collecting and evaluating bull genetic information. Then a methodology section reviews the FDH technique in the context of genetic traits as outputs. Results are presented followed by a summary and conclusion. Much empirical work on efficiency analysis entails the use of confidential individual firm or decision making unit data, restricting the publication of individual efficiency results. Because these dairy bull data are publically available, individual bull efficiency results can be disseminated to potential semen buyers.

## Data

Traditionally bulls were evaluated using information about their parents, and after they produced daughters, the characteristic and productivity of those daughters were observed and measured. These evaluations are called predicted transmitting abilities (PTA). However, transmitted traits of importance can now be genetically identified as soon as the bull is born and are called genomic predicted transmitting abilities (GPTA),[16]. This leads to determining transmittable traits by genetically testing with further evaluation based upon the performance of daughters [17]. The value of those individual PTA traits are determined based upon the USDA calculated economic value of each trait [5]. For instance, two important traits measured are the pounds of protein and the pounds of fat that a daughter will produce. These are very important traits because farmers are often paid not on the volume of milk produced, but rather on the pounds of protein and the pounds of fat found in the milk. Protein is important for cheese production and fat is importance for ice cream production. The price or value of either protein or fat is the net return from one additional pound produced from a representative cow. Net return per pound is the price of protein or fat minus the cost of additional feed required to produce an additional pound of protein or fat, assuming that other costs remain constant with a one pound increase in protein or fat. For a disease trait, such as resistance to mastitis, the additional net cost of mastitis is computed independent of the impact that mastitis may have on the other traits. That would essentially be the cost of treatment, which is the cost of any microbial use and the value of lost milk that is discarded during the treatment period [5].

We obtained bull genomic trait information and semen prices from the National Association of Animal Breeder certified semen services. This dataset consisted of 407 active Artificial Insemination (AI) Holstein bulls at the time the data were assessed (October 27, 2017). These data are generated by genetic testing of new bulls before they are used for breeding, with updates from the performance of their daughters [17]. In addition to pounds of protein (P) and pounds of fat (F), the other valuable output traits included in our analysis were somatic cell count (SCC), calving ability (CA), daughter pregnancy rate (DPR), udder composition (UC), and livability (LV). These are important traits in milk production. Disease traits were recently added after the data were assessed and is not included in the analysis. The summary statistics are listed in Table 1.

Most of the traits reported in the published dataset are centered on zero by subtracting the value of the trait of an average animal. This gives farmers a way of comparing an animal as being better or worse than average. We are using the FDH model where all input and output quantity variables must be non-negative. To produce positive values for all output traits, the trait for the base average animals for each attribute is added to each value of that observed

**Table 1. Bull data summary statistics from the list of active AI Holstein dairy bulls available in the United States as of October 27, 2017.**

| Variable | Mean | Standard Deviation | Minimum | Maximum |
|---|---|---|---|---|
| Pounds of protein | 849 | 20.36 | 751 | 910 |
| Pounds of fat | 1040 | 27.82 | 935 | 1115 |
| Somatic cell count | 0.61 | 0.17 | 0.01 | 1.00 |
| Calving ability | 95.24 | 22.63 | 0.10 | 150.20 |
| Daughter pregnancy rate | 9.41 | 1.89 | 3.9 | 16.1 |
| Udder composition | 4.22 | 0.85 | 0.01 | 6.55 |
| Livability | 85.50 | 2.19 | 78.90 | 91.60 |
| Bull semen price | 15.25 | 6.78 | 2.00 | 32.00 |

variable. The converted trait is then positive, with the lowest value of a trait being zero. Udder composition and calving ability are two attributes that are not indexed to a base animal value, but can have negative values, so we took the most negative value of each of these traits and added that value to all of the observations for that trait, making all of the observations nonnegative.

The somatic cell count (SCC) score is a measure of milk quality and the higher the value the lower the milk quality. To convert this score so that higher is better, we subtracted the maximum somatic cell score value found in the dataset from each bull's observed somatic cell score value. Then the absolute value of the transformation was used in order to change the sign from negative to positive. This changes the somatic cell score variable so that now a higher score is preferred.

The bull dataset does not come with a calculated calving ability (CA$) index variable. We were able to create this index using the method of [18], because the data did have the four components used to construct this index. CA$ is estimated by aggregating daughter stillbirths (DSB), service sire stillbirths (SSB), daughter calving ease (DCE), and service sire calving ease (SCE).

## Calculating efficiencies using free disposal hull

Although the implementation of convexity in production is often questioned and debated[19] and [20], in this application it is obvious that because only one sperm fertilizes the ovum, linear combination of bulls cannot occur, so a FDH specification was used which relaxes the convexity assumption of the Data Envelopment Analysis (DEA) model. An output-oriented method rather than an input-oriented method was used because we are measuring by how much output the characteristics of bulls can be proportionally expanded without changing the level of a given input, which in this case is the bull semen and is fixed.

When the convexity assumption is relaxed, the efficient frontier looks like a staircase pattern consisting of linear segments. Fig 1 shows the graphical representation of technical, revenue, and allocative efficiency. The dotted stair-step line is the FDH frontier of interest, the smooth black line represents a DEA frontier. Line P1/P2 is the price ratio for the two outputs (protein and fat), which determines a bull's revenue efficiency. We estimate both technical and revenue efficiency and then decompose those estimates to find allocative efficiency, because calculated revenue efficiency (RE) divided by technical efficiency (TE) derives allocative efficiency (AE).

In Fig 1, bulls B, C and D would be labeled as technically efficient, while bull A is technically inefficient because it is below the production possibility curve defined by the FDH frontier. The distance between the points A and E represent the technical inefficiency of bull A.

In Fig 1, the efficiencies of bull A are: TE = (0A)/(0E) and AE = (0E)/(0F). Revenue efficiency can be decomposed by RE = (TE)*(AE) = (0A)/(0E)*(0E)/(0F) = (0A)/(0F). This last equation can be transformed to find allocative efficiency as AE = (RE)/(TE) = (0E)/(0F).

To empirically calculate technical efficiency (TE) using the FDH model, the output measure can be estimated with the binary programming model, where $i = 1,\ldots, n$ is the number of traits:

$$\hat{\lambda}_{FDH}(x_0, y_0) = \max\{\lambda | \lambda y_0 \le \sum_{i=1}^{n} \gamma_i Y_i; \ x_0 \ge \sum_{i=1}^{n} \gamma_i X_i, \sum_{i=1}^{n} \gamma_i = 1;$$
$$\gamma_i \in \{0,1\}, i = 1, \ldots, n\}$$

The equation above produces a point estimate for each bull. To get a variation around that point estimate the bootstrapping method of [21] was used.

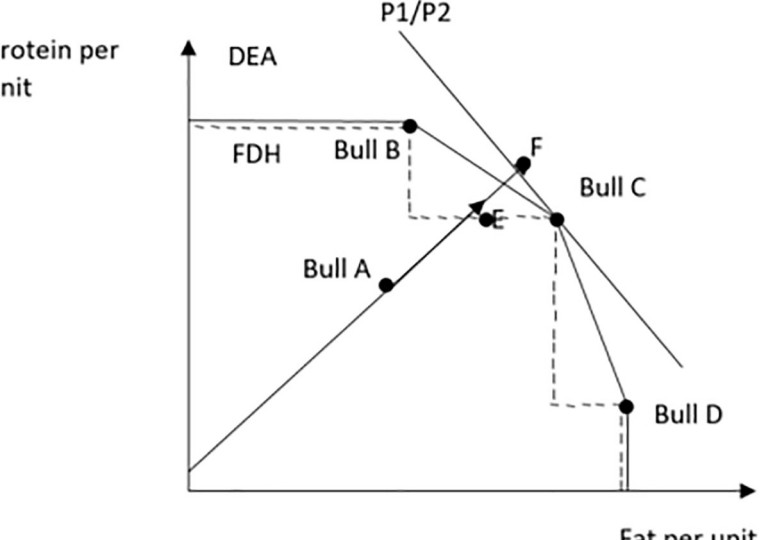

**Fig 1. Graphical representation of allocative efficiency for two outputs (protein and fat) and one input (semen) case.**

Revenue efficiency (RE) can be calculated by first multiplying the output prices by their corresponding quantity attribute using the USDA economic returns from each trait as the set of output prices. The maximum revenue of the bull possessing the maximum revenue was used as the divisor for each bull revenue (summed net merit) to calculate revenue efficiency of each bull.

Profit efficiency was calculated similarly as revenue efficiency, where the bull with the highest profit was used as the dominator to each bull's profit. We also determine the semen price necessary for each bull to be profit efficient.

These techniques to estimate bull efficiencies can also be applied using the genetics of the female bovine to help determine which heifers to retain for the herd before the cost of raising those animal are incurred, and to determine which cows might be culled.

## Results

Because farmers are paid based upon the pounds of protein and the pounds of fat that a cow produces in her milk, and the other attribute traits indirectly affects profit, mostly through

**Table 2. Summary bull efficiency estimates from free disposal hull analysis (fat and protein as outputs, semen as input).**

|  | Mean | Range |
| --- | --- | --- |
| Technical efficiency | 0.94 (0.02)* (.0113)^ | {1.0, .84} |
| Revenue efficiency | 0.94 (0.02)* | {1.0, .84} |
| Allocative efficiency | 0.99 (0.004)* | {1.0, .98} |
| Profit efficiency | 0.94 (0.02)* | {1.0, .84} |

*Standard deviations in parentheses

^ Simar and Wilson bootstrapping estimate of mean variability

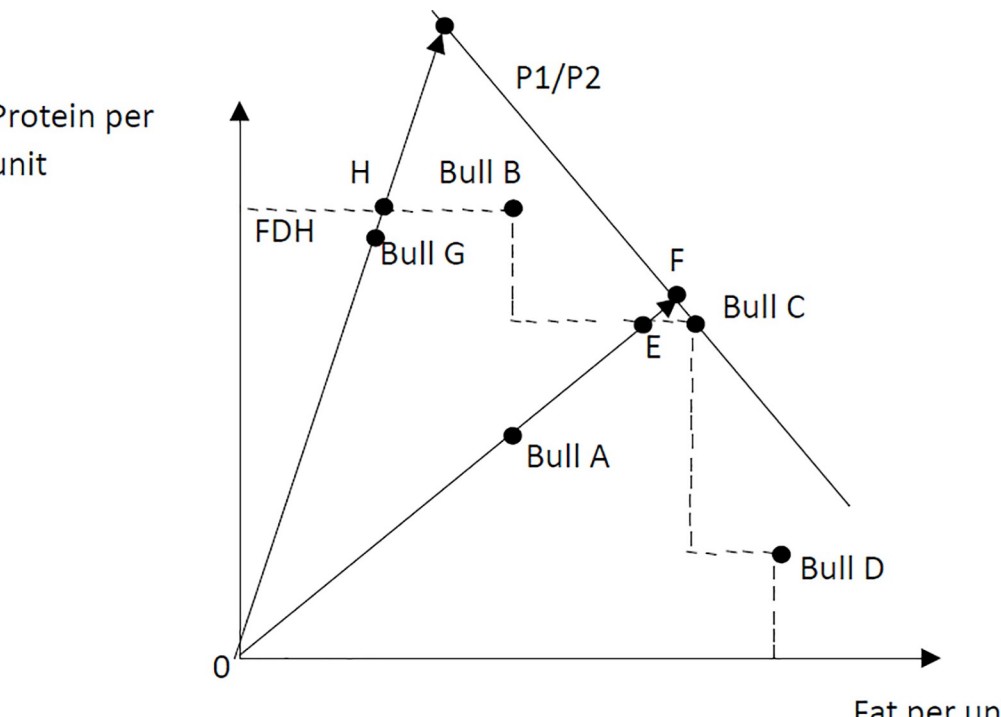

**Fig 2. Graphical representation of Free Disposal Hull (FDH) with two outputs (protein and fat) and one input (semen).**

cost impacts, our base analysis was done with protein and fat as the only two outputs and the sole input of the semen. Table 2 shows the summary statistics for technical, allocative, revenue, and profit efficiency. The mean technical efficiency is .94, with a minimum score of .84. Two bulls are technically efficient, with one bull producing more protein and the other bull producing more fat such that they define the efficient technical frontier with two outputs. All other bulls produce lower quantities of protein and fat than these two bulls defining the frontier. Revenue efficiency has a mean score of .94 and a minimum score of .84. Of the four efficiency measures, allocative efficiency has the highest mean value score of .99. Thus, many of these bulls are producing the correct or close to the correct proportion of outputs based on the prices used for protein and fat, they are simply less efficient technically in producing those outputs. Only one bull is allocative efficient, and that bull is also the sole bull that is profit efficient. Profit efficiency has a mean value score of .94, which is equal to the revenue efficiency score of .94. However, when additional outputs are added later, average profit efficiency becomes higher than average revenue efficiency, implying that semen is at least partially priced to offset lower revenue, i.e. bulls with higher revenue efficiency have semen that is priced higher.

A bull with a high technical efficiency score generally has a lower allocative efficiency score. The correlation between the allocative efficiency and technical efficiency was negative 0.07. Fig 2 shows the graphical representation of FDH with one input and two outputs to explain how technical and allocative efficiency can have a negative relationship. From Fig 2, it is evident that bull A and bull G are not technical, revenue, allocative or profit efficient. For bull A, TE = 0A/0E, AE = 0E/0F, and RE = 0A/0F. The distance EF is much smaller than the distance AE, meaning bull A will have a much higher allocative efficiency score than a technical efficiency score. For bull G, TE = 0G/0H, AE = 0H/0I, and RE = 0G/0I. The distance GH is smaller than the distance HI, making this bull more technical efficient than allocative efficient.

Technical efficiency and revenue efficiency generally are positively correlated, because technical efficiency is embedded within revenue efficiency. This means bulls with high technical efficiency scores will generally also have high revenue efficiency scores. The correlation between revenue efficiency, which is synonymous with the USDA Net Merit Value (NM$), and profit efficiency with just the outputs of protein and fat is high at 0.99. This implies that the market for semen, involving the interaction of farmers selecting semen and providers pricing semen, may be strongly driven by the protein and fat traits. This is reasonable given that the milk check is directly dependent upon these two outputs, while the other traits indirectly impact the milk check.

When somatic cell score was added as a third output to the model, the average revenue efficiency remained constant at .94, but allocative efficiency decreased. Technical efficiency is embedded in revenue efficiency, and technical efficiency increased when additional outputs are added to the model. However, allocative efficiency decreased because more outputs allow a greater chance a bull is not producing the correct mix of outputs given the assumed prices of the output traits.

In this three output, one input model, 22 bulls are technically efficient; however, only one bull is allocative and profit efficient. Table 3 shows the summary statistics for technical, allocative, revenue, and profit efficiency. Most of the other bull's efficiency scores decrease moving from technical to allocative efficiency. When semen prices are added into the analysis, profit efficiency remained at .99. A kernel distribution of the histogram of the four efficiencies are shown in Fig 3. When SCC is added and there is only one input, it is evident that making decisions based on technical efficiency alone is not sufficient for picking a bull. The distribution clearly shows that many bulls that are highly technically efficient are not revenue or profit efficiency.

When the additional four traits (seven total) are added sequentially to the model, empirical results change, but generally follow an expected pattern. Technical efficiencies increase with each additional trait added because of the occurrence that a bull may excel in the added trait. This phenomenon is not unique to this data and occurs in technical efficiency analysis [22]. In contrast, revenue efficiency generally falls because, although a bull may have a high technical efficiency with the larger set of traits, it is not producing the most valuable mix of traits given the prices used to value those traits. As more traits are added to the empirical analysis, the more likely the bull is not producing the highest revenue traits. Because allocative efficiency is calculated as revenue efficiency divided by technical efficiency, allocate efficiency also generally decreases with the addition of more traits unless the bull is completely technical efficient,

**Table 3. Summary bull efficiency estimates from free disposal hull analysis (fat, protein and somatic cell count as outputs, semen as sole input).**

|  | Mean | Range of Estimates |
|---|---|---|
| Technical efficiency | 0.96<br>(0.02)*<br>(0.01)^ | {1.0, .84} |
| Revenue efficiency | 0.94<br>(0.02)* | {1.0, .83} |
| Allocative efficiency | 0.98<br>(0.01)* | {1.0, .94} |
| Profit efficiency | 0.94<br>(0.02)* | {1.0, .84} |

*Standard deviations in parentheses

^ Simar and Wilson bootstrapping estimate of mean variability

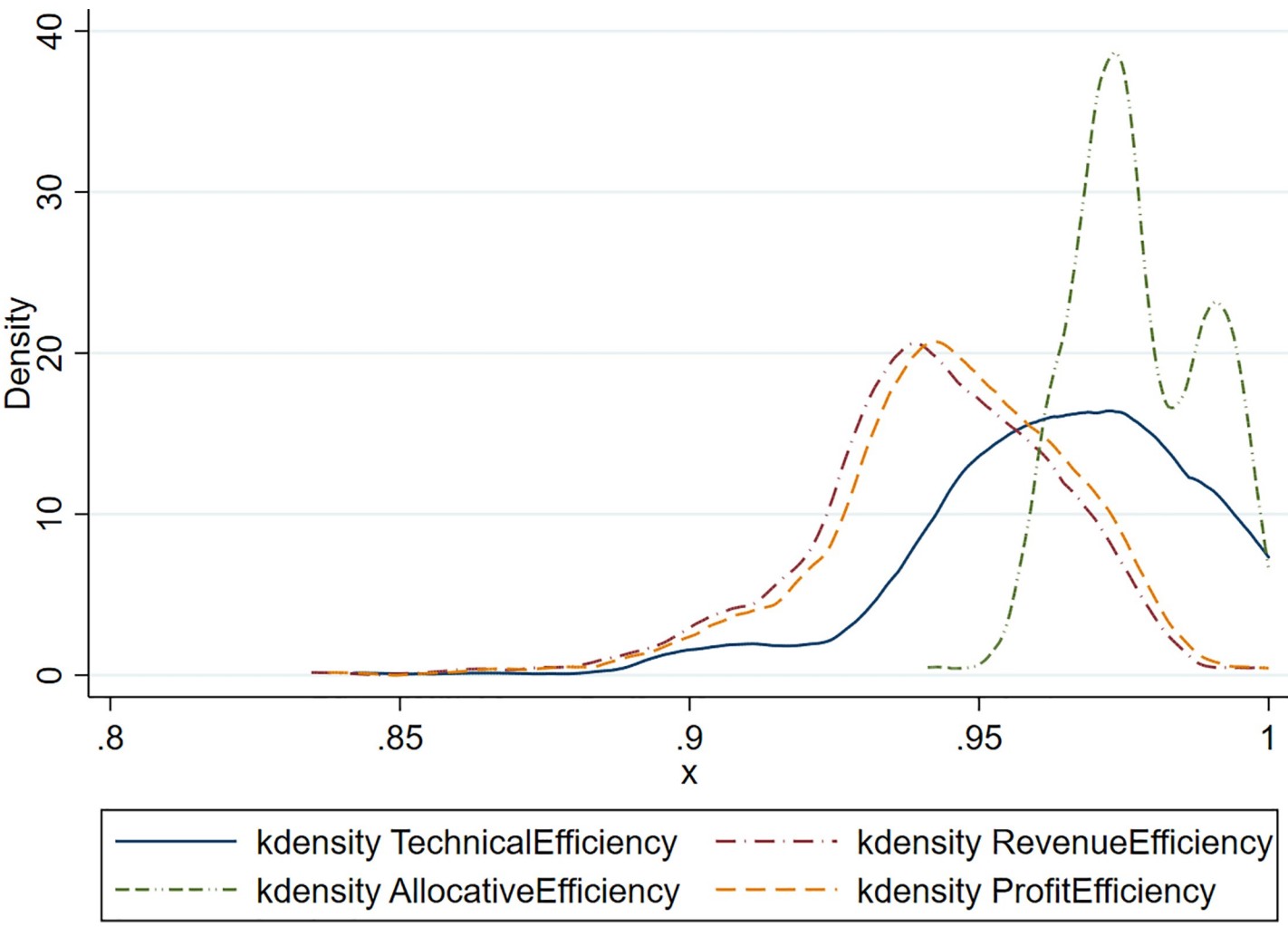

**Fig 3. Kernel density estimation of efficiencies including fat, protein and somatic cell count as outputs in FDH model.**

in which case allocative efficiency is identical to revenue efficiency. The correlation between technical efficiency and allocative efficiency with this seven-output model is 0.20. Profit efficiency generally falls because of the decrease in revenue efficiency. Yet, the correlation between revenue efficiency and profit efficiency is still high at 0.99.

The results with all seven traits as outputs are summarized in Table 4 with histograms of each efficiency measure shown in Fig 4. The technical efficiency histogram clearly shows that almost all bulls are technically efficient. The distribution of allocative efficiency becomes unimodal and shifts to the left compared to the previous distribution with only the protein, fat and SCC traits. The distributions of revenue efficiency and profit efficiency also shifts to the left, but become almost identical because the only difference between these two measures is the subtraction of the semen price from revenue. The fact, however, that the profit efficiency distribution lies slightly to the right of the revenue efficiency provide evidence that some semen is not correctly priced.

Although separate results for all seven models when traits are sequentially added are not presented in detail, there was commonality across models with a high degree of correlation of the efficiency of individual bulls. Table 5 contains the correlations for technical efficiencies

**Table 4. Summary bull efficiency estimates from free disposal hull analysis (fat, protein, somatic cell count, liveability, udder composition, and daughter pregnancy rate and calving ability as outputs, semen as sole input).**

|  | Mean | Range of Estimates |
|---|---|---|
| Technical efficiency | 0.99 (0.02)* (0.02)^ | {1.0, .91} |
| Revenue efficiency | 0.95 (0.02)* | {1.0, .84} |
| Allocative efficiency | 0.96 (0.01)* | {1.0, .92} |
| Profit efficiency | 0.95 (0.02)* | {1.0, .84} |

*Standard deviations in parentheses

^ Simar and Wilson bootstrapping estimate of mean variability

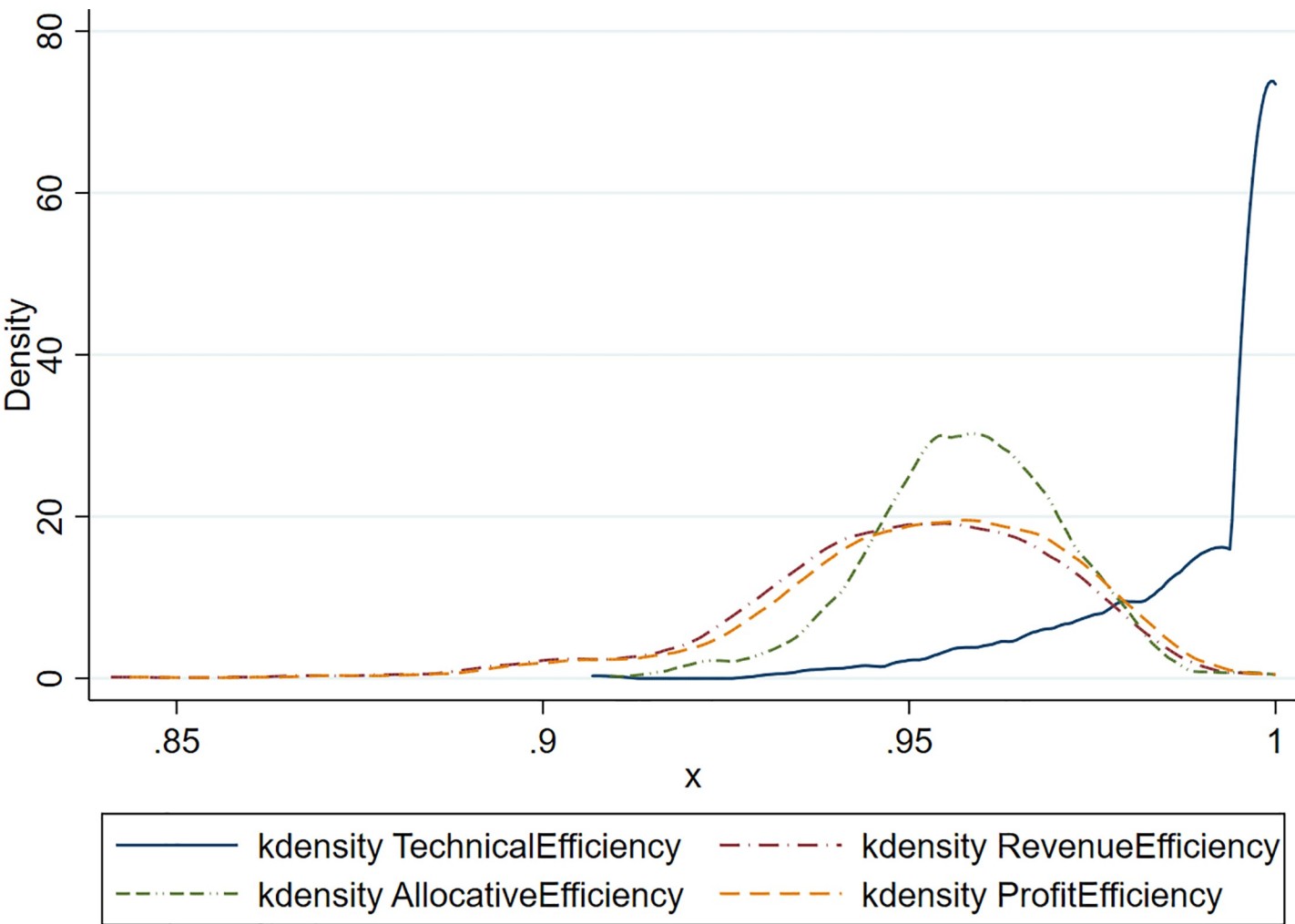

**Fig 4. Kernel density estimation of efficiencies including fat, protein, somatic cell count, liveability, udder composition, daughter pregnancy rate and calving ability as outputs in FDH model.**

**Table 5. Correlations for technical efficiency for bulls across models using various outputs.**

| | Fat, protein | Fat, protein, SCC | Fat, protein, CA$ | Fat, protein, SCC, CA$ | Fat, protein, SCC, CA$, DPR | Fat, protein, SCC, CA$, DPR, UC | Fat, protein, SCC, CA$, DPR, UC, LV |
|---|---|---|---|---|---|---|---|
| Fat, protein | 1 | | | | | | |
| Fat, protein, SCC | .82 | 1 | | | | | |
| Fat, protein, CA$ | .94 | .80 | 1 | | | | |
| Fat, protein, SCC, CA$ | .72 | .94 | .77 | 1 | | | |
| Fat, protein, SCC, CA$, DPR | .73 | .88 | .78 | .92 | 1 | | |
| Fat, protein, SCC, CA$, DPR, UC | .68 | .81 | .72 | .85 | .90 | 1 | |
| Fat, protein, SCC, CA$, DPR, UC, LV | .64 | .75 | .69 | .79 | .83 | .92 | 1 |

SCC = somatic cell count, CA$ = calving ability, DPR = daughter pregnancy rate, UC = udder composition, and LV = liveability

across the seven models. The weakest correlation is between the two output, one input and seven output one input models with a correlation of .64. The three output, one input model with fat, protein, and SCC, and the four output, one input models with fat, protein, SCS, and CA$ have the strongest correlation of .94. On average correlations for technical efficiency between models decreases as more outputs are included in the model. The fact that more bulls become completely technically efficient when more outputs are included in the model means there is a strong correlation between adjacent models.

Table 6 shows correlations for revenue efficiency between models. The most robust correlation across revenue efficiencies is .99 and occurs between the four outputs, one input, and five output adding DPR, one input models. Overall, the correlations for revenue efficiency for each model are extremely high, meaning no significant differences were found between the models. This implies that if a bull has a high revenue efficiency in the two output, one input model, it is very likely that bull also has a high revenue efficiency score in the seven output, one input model.

After examining the profit efficiencies for each model it is evident the some of the bulls are not profit efficient. Fig 5 is a scatter plot of semen price versus profit efficiency and shows only a slight positive relationship between these two variables. In order for those bulls to become profit efficient, their semen price must be reduced by the difference between the maximum

**Table 6. Correlations for revenue efficiency for bulls across models using various outputs.**

| | Fat, protein | Fat, protein, SCC | Fat, protein, CA$ | Fat, protein, SCC, CA$ | Fat, protein, SCC, CA$, DPR | Fat, protein, SCC, CA$, DPR, UC | Fat, protein, SCC, CA$, DPR, UC, LV |
|---|---|---|---|---|---|---|---|
| Fat, protein | 1 | | | | | | |
| Fat, protein, SCC | .99 | 1 | | | | | |
| Fat, protein, CA$ | .99 | .99 | 1 | | | | |
| Fat, protein, SCC, CA$ | .98 | .99 | .99 | 1 | | | |
| Fat, protein, SCC, CA$, DPR | .97 | .98 | .98 | .99 | 1 | | |
| Fat, protein, SCC, CA$, DPR, UC | .96 | .97 | .97 | .99 | .99 | 1 | |
| Fat, protein, SCC, CA$, DPR, UC, LV | .94 | .96 | .96 | .97 | .99 | .99 | 1 |

SCC = somatic cell count, CA$ = calving ability, DPR = daughter pregnancy rate, UC = udder composition, and LV = liveability

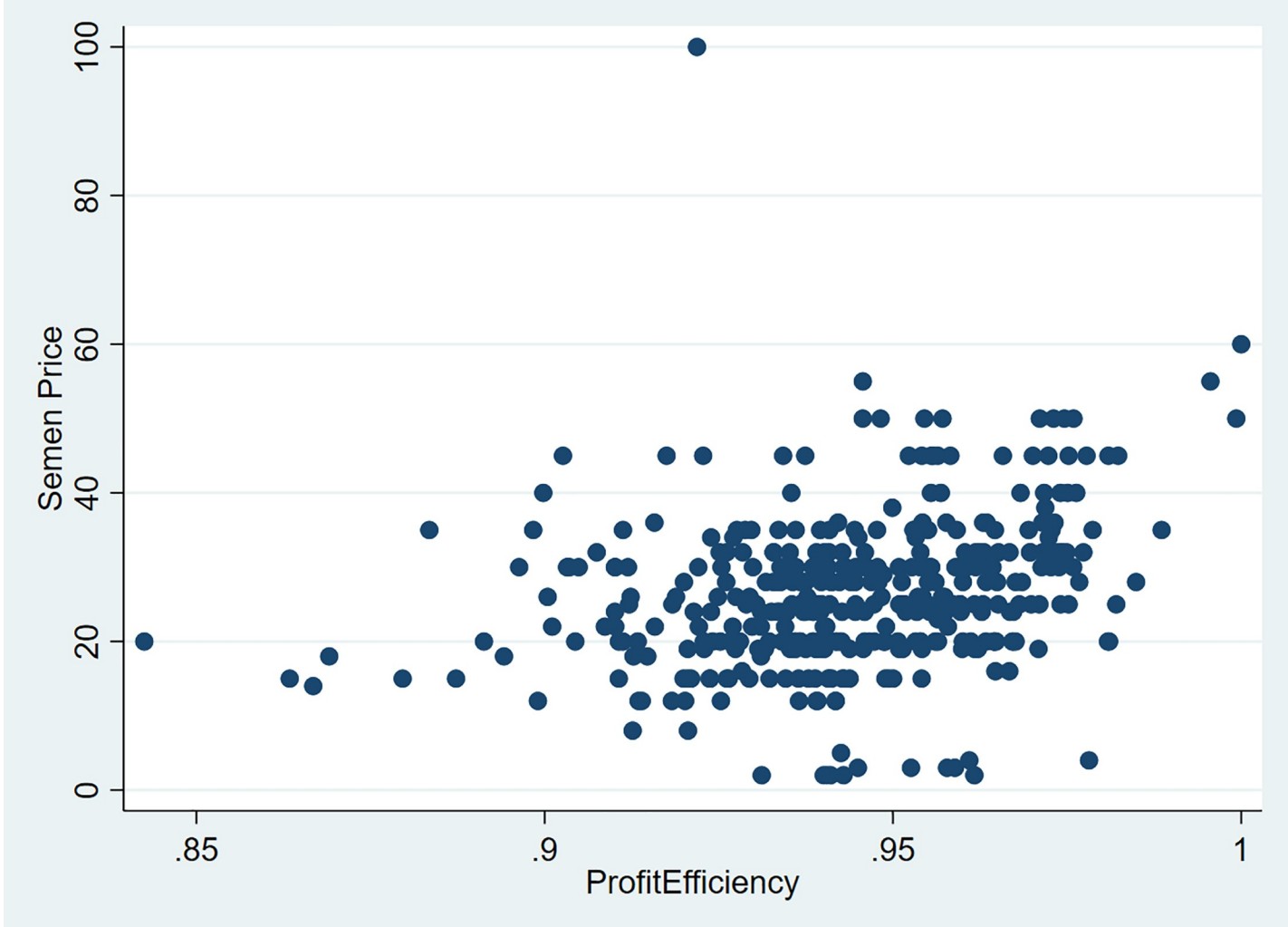

**Fig 5. Scatter plot of semen price versus profit efficiency from the two output (fat, protein), one input (semen) model.**

profit and actual profit. The two bulls that are close to being profit efficient are priced higher than the majority of the other bulls. Fig 6 is a histogram that shows the percent reduction in semen price required to make the bulls profit efficiency for the two output, one input model, and it is clear that over half the bulls need at least a 50% reduction in price in order to become profit efficient. A couple of bulls need their price reduced by over 100%, with the highest being around 250%.

## Summary and conclusions

Technical, revenue, allocative, and profit efficiencies were calculated for Holstein bulls using FDH models, sequentially adding seven genetic merits as outputs. Using data from the National Association of Animal Breeder certified semen services, the seven output traits specified were pounds of protein, pounds of fat, somatic cell count, calving ability, daughter pregnancy rate, udder composition, and livability. The USDA estimated economic value of each trait was used as the price for each genetic merit. The single input was the semen.

As more trait outputs are included in the model, more bulls become unique, increasing their technical efficiency. Allocative efficiency had the opposite result. Each time a new trait

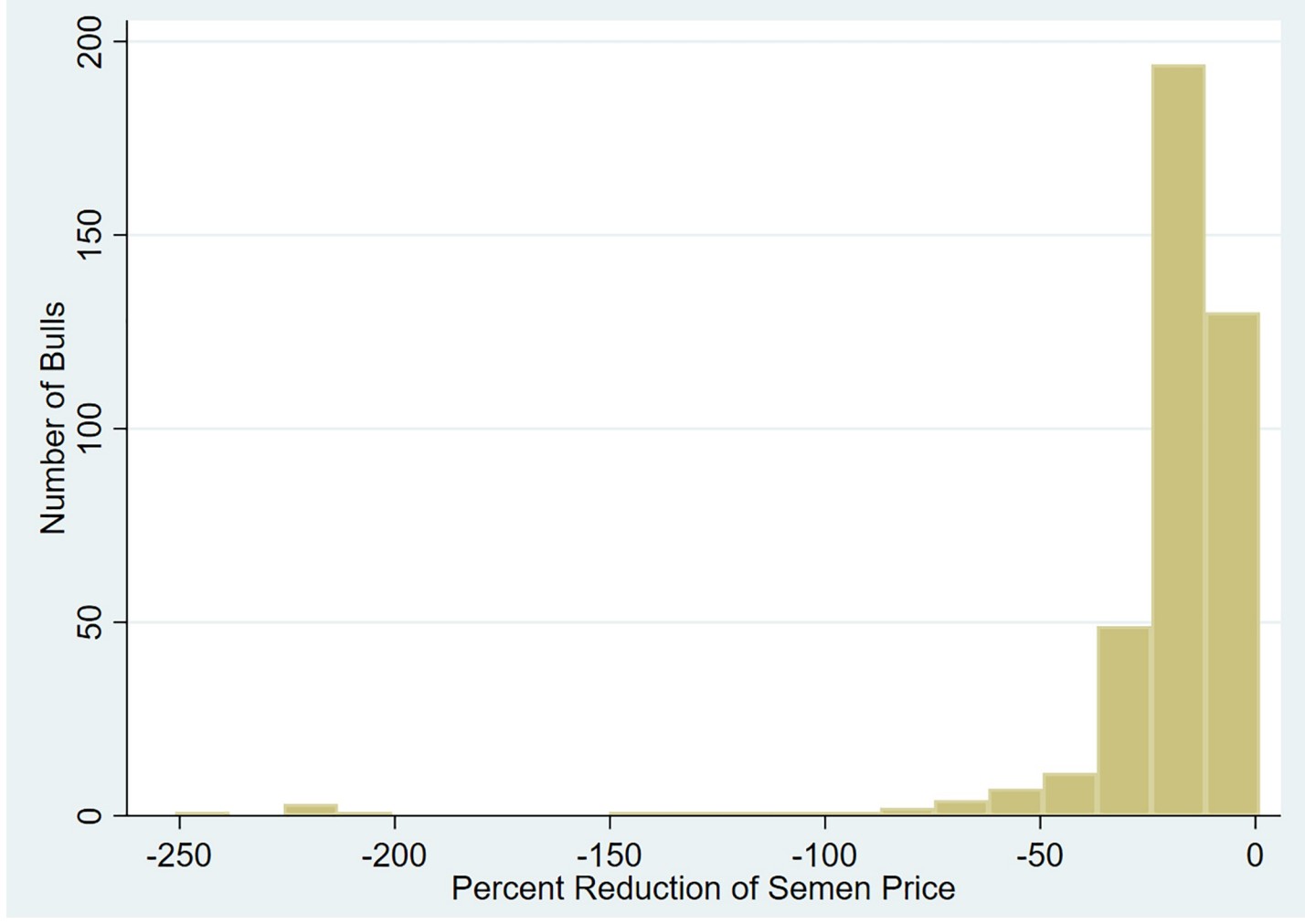

**Fig 6. Distribution of percent reduction of semen price necessary for profit efficiency (Two output (protein and butterfat), one input (semen) model).**

was included in the model, the mean allocative efficiency score decreased. Bulls are generally highly technical efficient but many are producing the wrong traits given the values of those traits. Other than fat and protein, most bulls are not producing the correct proportion of genetic outputs given trait values. Thus generally most bulls are highly technical efficient but only a few are allocative, profit efficient, and revenue efficient.

Revenue and profit efficiency scores overall increased with additional traits added to the model, but not as dramatically as technical efficiency. Revenue efficiency results were similar to profit efficiency results, differing only from the addition of the semen price, and distributions across bulls were similar, but with profit efficiency lower.

Many bulls were not profit efficient, implying they were overpriced given the value of their traits. For the model of only the two outputs of protein and butterfat, and one input, the percent reduction in which the semen price would need to be reduced was estimated for a bull to be profit efficient. The results were quite staggering considering a couple of the semen prices would need to be reduced by over 100%; however, the value of the traits may not be solely additive with future analyses

The Net Merit Value (NM$) produced by the USDA has served farmers well in selecting appropriate bulls and making significant progress in the genetics of the herds. The revenue efficiency technique presented here is synonymous with NM$ but with an index value between zero to one, with the value of one representing complete revenue efficient. Given that revenue efficiency was quite high in our analysis strongly supports the use of the USDA NM$ in bull selection. However, the breakdown of revenue efficiency into the product of technical efficiency and allocative efficiency, again indexed from zero to one, provides more details for bull selection. The addition of profit efficiency incorporates the cost of the semen into the efficiency calculation and ultimately the efficiency index farmers should use in their semen selection decision.

New bulls are added continuously to the list of available bulls and computed efficiencies could change when new bull lists are released. Updated efficiencies could be calculated and published. Younger bulls with improved genetics may cause older bulls to decrease in efficiency. Individual farmers also receive protein and fat prices different from averages, and the implicit value they place on other traits may differ from those used in the USDA data. With the data connected dairy farm, individual farms can perform their unique analysis using their specific prices and trait values. As farms increase their genetic testing of cows and potential heifers, these models can also be applied to assess the efficiency of those individual animals in their herds. The efficiencies estimated would be in reference to their top animals and would assist in moving average herd genetics to that frontier. These farmers could even add the available performance data of cows outside their herd to the reference set to assess the genetic efficiency of their herd relative to the industry.

## Acknowledgments

The findings and conclusions in this publication are those of the author and should not be construed to represent any official USDA or U.S. Government determination or policy.

## Author Contributions

**Writing – original draft:** Christine E. Whitt.

**Writing – review & editing:** Loren W. Tauer, Heather Huson.

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
