## [Decision Letter · Decision Letter 0]

11 Aug 2019

PONE-D-19-18307

Efficiency of Dairy Genetic Traits

PLOS ONE

Dear Mrs. Whitt,

Thank you for submitting your manuscript to PLOS ONE. After careful consideration, we feel that it has merit but does not fully meet PLOS ONE’s publication criteria as it currently stands. Therefore, we invite you to submit a revised version of the manuscript that addresses the points raised during the review process.

We would appreciate receiving your revised manuscript by Sep 25 2019 11:59PM. To enhance the reproducibility of your results, we recommend that if applicable you deposit your laboratory protocols in protocols.io, where a protocol can be assigned its own identifier (DOI) such that it can be cited independently in the future. For instructions see: http://journals.plos.org/plosone/s/submission-guidelines#loc-laboratory-protocols

We look forward to receiving your revised manuscript.

Kind regards,

Juan J Loor

Academic Editor

PLOS ONE

Journal Requirements:

Reviewers' comments:

Reviewer's Responses to Questions

**Comments to the Author**

1. Is the manuscript technically sound, and do the data support the conclusions?

Reviewer #1: Partly

2. Has the statistical analysis been performed appropriately and rigorously? 

Reviewer #1: No

3. Have the authors made all data underlying the findings in their manuscript fully available?

Reviewer #1: Yes

4. Is the manuscript presented in an intelligible fashion and written in standard English?

Reviewer #1: No

5. Review Comments to the Author

Reviewer #1: 1. Manuscript is technically sound and data supports the conclusions. I answered 'partly' to this question because greater rigor would have looked at the effects over time. The current study is one time point and additional time points of bull data could have strengthen the generality of conclusions. Also,a linear combination of bulls could be appropriate if one evaluated a herd of cows. The Free Disposal Hull approach would not be needed for bull selection in a herd.

2. My criticism here is not of the statistical analysis, but of the reporting of the results. Four decimals is more than can be supported based on the standard deviations or sample size of approximately 400 bulls.

3. Data is available.

4. Most of the manuscript is presented well. A few sentences are missing words. Search for 'form' and replace with 'farm' in at least one place. 'Since' should often be replaced with 'Because'. Since implies time, which is not how it was used in most places. In many of the sentences with 'there are or there were', a revision of the sentences would improve readability.

5. If my name will be associated with this review, I will advertise that I analyzed a similar data set with linear programing a few decades ago. (Shanks, R. D. and A. E. Freeman. 1979. Choosing progeny-tested Holstein sires that meet genetic goals at minimum semen costs. J. Dairy Sci. 62:1429-1434.) The market does a good job of arriving at a reasonable semen price for most bulls. When discrepancies occur, often some information not included in the data set is responsible for the outliers.

6. PLOS authors have the option to publish the peer review history of their article (what does this mean?). If published, this will include your full peer review and any attached files.

Reviewer #1: Yes: Roger D. Shanks

---

## [Author Response · Author response to Decision Letter 0]

19 Aug 2019

1. Manuscript is technically sound and data supports the conclusions. I answered 'partly' to this question because greater rigor would have looked at the effects over time. The current study is one time point and additional time points of bull data could have strengthen the generality of conclusions. Also, a linear combination of bulls could be appropriate if one evaluated a herd of cows. The Free Disposal Hull approach would not be needed for bull selection in a herd.

We agree that analysis over time with updated bull lists would be an interesting application. In our summary, we state that analysis with an updated bull list could result in current bulls experiencing a decrease in calculated efficiencies as new bulls are added to the list. That should be expected with genetic progress. We leave that analysis for another paper.

Models that investigate bull selection given the current traits in the herd with the goal of increasing certain traits could utilize linear programming to maximize the value of the herd. Our analysis simple looks at the traits of a single bull with a comparison to every other bull, thus the use of the Free Disposal Hull technique. 

2. My criticism here is not of the statistical analysis, but of the reporting of the results. Four decimals is more than can be supported based on the standard deviations or sample size of approximately 400 bulls.

 We have made the recommended change. Now we are reporting our results with only two decimals instead of four. 

3. Data is available.

The data is publicly available like you mentioned 

4. Most of the manuscript is presented well. A few sentences are missing words. Search for 'form' and replace with 'farm' in at least one place. 'Since' should often be replaced with 'Because'. Since implies time, which is not how it was used in most places. In many of the sentences with 'there are or there were', a revision of the sentences would improve readability.

 We searched the manuscript and corrected the words, ‘form to ‘farm’, and changed ‘since to because’ as you recommend and did a through read of the paper and made a few other editing changes to improve the readability of the manuscript. 

5. If my name will be associated with this review, I will advertise that I analyzed a similar data set with linear programing a few decades ago. (Shanks, R. D. and A. E. Freeman. 1979. Choosing progeny-tested Holstein sires that meet genetic goals at minimum semen costs. J. Dairy Sci. 62:1429-1434.) The market does a good job of arriving at a reasonable semen price for most bulls. When discrepancies occur, often some information not included in the data set is responsible for the outliers.

Thank you for your suggestion. We added a sentence in the text citing the paper.

---

## [Decision Letter · Decision Letter 1]

9 Sep 2019

PONE-D-19-18307R1

Production Efficiency of Dairy Genetic Traits

PLOS ONE

Dear Mrs. Whitt,

Thank you for submitting your manuscript to PLOS ONE. After careful consideration, we feel that it has merit but does not fully meet PLOS ONE’s publication criteria as it currently stands. Therefore, we invite you to submit a revised version of the manuscript that addresses the points raised during the review process.

We would appreciate receiving your revised manuscript by Oct 24 2019 11:59PM. To enhance the reproducibility of your results, we recommend that if applicable you deposit your laboratory protocols in protocols.io, where a protocol can be assigned its own identifier (DOI) such that it can be cited independently in the future. For instructions see: http://journals.plos.org/plosone/s/submission-guidelines#loc-laboratory-protocols

We look forward to receiving your revised manuscript.

Kind regards,

Juan J Loor

Academic Editor

PLOS ONE

Reviewers' comments:

Reviewer's Responses to Questions

**Comments to the Author**

1. If the authors have adequately addressed your comments raised in a previous round of review and you feel that this manuscript is now acceptable for publication, you may indicate that here to bypass the “Comments to the Author” section, enter your conflict of interest statement in the “Confidential to Editor” section, and submit your "Accept" recommendation.

Reviewer #1: (No Response)

2. Is the manuscript technically sound, and do the data support the conclusions?

Reviewer #1: Partly

3. Has the statistical analysis been performed appropriately and rigorously? 

Reviewer #1: Yes

4. Have the authors made all data underlying the findings in their manuscript fully available?

Reviewer #1: Yes

5. Is the manuscript presented in an intelligible fashion and written in standard English?

Reviewer #1: Yes

6. Review Comments to the Author

Reviewer #1: Christine, Loren and Heather,

Thank you for incorporating my previous comments. It was not necessary to cite my previous paper on a related topic, but your gesture was appreciated.

On the revision, I was surprised by the title change. Production efficiency is not quite right for me. Within the text, you do mention bull efficiency. When I first saw the revision, I jotted down bull use efficiency. I also tried economic efficiency in the title. Because you do use the phrase in the text, please consider "Bull efficiency of dairy genetic traits" for your title.

A few line specific comments follow (lines are from tracked changes version):

line 69, lines 106 to 107, lines 172 to 173, line 359: How many times does Free Disposal Hull need to be defined as FDH? Unfortunately, I did not look up a PLOS ONE policy for abbreviations. Also, can an abbreviation start a sentence as in line 69?

line 72: Consider replacing "efficiency" with "efficient" in this context.

line 116: "predicted transmitting abilities (PTA)

line 117 to 118: "genomic predicted transmitting abilities (GPTA)

line 196: Is n the number of traits in this application?

line 301 to 303: Profit efficiency looks to be slightly to the right of revenue efficiency distribution to me on Figure 4. Are labels correct?

line 348: Delete extra "to need".

lines 376 to 380: Interesting interpretation. I am not suggesting any changes, but consider that value of traits may not be solely additive with future analyses.

Figure 1: If possible include a few more definitions for this figure. What is P1/P2? Is line AF perpendicular to P1/P2 line? From figure 2, I surmise that line AF should go through zero, zero and not necessarily be perpendicular. My guess is that my questions are pretty basic for the method, but I did not take the time to find the FDH reference.

Interesting paper.

Thanks for taking a look at this topic.

Have a great day.

Roger Shanks

7. PLOS authors have the option to publish the peer review history of their article (what does this mean?). If published, this will include your full peer review and any attached files.

Reviewer #1: Yes: Roger D. Shanks

---

## [Author Response · Author response to Decision Letter 1]

17 Sep 2019

You suggest the title, “Bull Efficiency of Dairy Genetic Traits” which is better than our title. However, we think that placing the word “bull” before the word “genetic” rather than “efficiency” is more correct and informative. Thus our new title is “Efficiency of Dairy Bull Genetic Traits”

line 69, lines 106 to 107, lines 172 to 173, line 359: How many times does Free Disposal Hull need to be defined as FDH? Unfortunately, I did not look up a PLOS ONE policy for abbreviations. Also, can an abbreviation start a sentence as in line 69?

I looked up the PLOS ONE policy for abbreviations and it states “Define abbreviations upon first appearance in the text.” I have edited the manuscript to fit the correct guidelines. 

line 72: Consider replacing "efficiency" with "efficient" in this context.

Replacement was made.

line 116: "predicted transmitting abilities (PTA)

Correction made.

line 117 to 118: "genomic predicted transmitting abilities (GPTA)

Corrections made.

line 196: Is n the number of traits in this application?

I added the wording, “where i=1,…, n is the number of traits:” before equation one to address this comment. 

line 301 to 303: Profit efficiency looks to be slightly to the right of revenue efficiency distribution to me on Figure 4. Are labels correct?

The labels are correct and I fixed my mistype from “left” to “right”. Thank you for finding this error. 

line 348: Delete extra "to need".

Deleted as suggested.

lines 376 to 380: Interesting interpretation. I am not suggesting any changes, but consider that value of traits may not be solely additive with future analyses.

I included an extra sentence stating your suggestion. 

Figure 1: If possible include a few more definitions for this figure. What is P1/P2? Is line AF perpendicular to P1/P2 line? From figure 2, I surmise that line AF should go through zero, zero and not necessarily be perpendicular. My guess is that my questions are pretty basic for the method, but I did not take the time to find the FDH reference.

I included an extra sentence at lines 181-182 where to elaborate on figure 1.

---

## [Editor Report · Decision Letter 2]

23 Sep 2019

Bull Efficiency Using Dairy Genetic Traits

PONE-D-19-18307R2

Dear Dr. Whitt,

We are pleased to inform you that your manuscript has been judged scientifically suitable for publication and will be formally accepted for publication once it complies with all outstanding technical requirements.

With kind regards,

Juan J Loor

Academic Editor

PLOS ONE
---

## [Editor Report · Acceptance letter]

28 Oct 2019

PONE-D-19-18307R2 

Bull Efficiency Using Dairy Genetic Traits 

Dear Dr. Whitt:

I am pleased to inform you that your manuscript has been deemed suitable for publication in PLOS ONE. Congratulations! Your manuscript is now with our production department. 

With kind regards,

on behalf of

Dr. Juan J Loor 

Academic Editor

PLOS ONE